# Transformer-Based Language Model Surprisal Predicts Human Reading Times Best with About Two Billion Training Tokens

**Byung-Doh Oh**
Department of Linguistics
The Ohio State University
oh.531@osu.edu

**William Schuler**
Department of Linguistics
The Ohio State University
schuler.77@osu.edu

## Abstract

Recent psycholinguistic studies have drawn conflicting conclusions about the relationship between the quality of a language model and the ability of its surprisal estimates to predict human reading times, which has been speculated to be due to the large gap in both the amount of training data and model capacity across studies. The current work aims to consolidate these findings by evaluating surprisal estimates from Transformer-based language model variants that vary systematically in the amount of training data and model capacity on their ability to predict human reading times. The results show that surprisal estimates from most variants with contemporary model capacities provide the best fit after seeing about two billion training tokens, after which they begin to diverge from humanlike expectations. Additionally, newly-trained smaller model variants reveal a 'tipping point' at convergence, after which the decrease in language model perplexity begins to result in poorer fits to human reading times. These results suggest that the massive amount of training data is mainly responsible for the poorer fit achieved by surprisal from larger pre-trained language models, and that a certain degree of model capacity is necessary for Transformer-based language models to capture humanlike expectations.

## 1 Introduction

The predictability of upcoming linguistic material has long been considered a crucial factor underlying difficulty in human sentence processing (Hale, 2001; Levy, 2008), and has received empirical support from numerous studies showing surprisal (Shannon, 1948) to be highly predictive of relevant behavioral and neural measures (e.g. Demberg and Keller, 2008; Smith and Levy, 2013; Hale et al., 2018; Shain et al., 2020). Since language models (LMs) are trained to estimate a conditional probability distribution of a word given its context, surprisal estimates calculated from them have often

been evaluated on their ability to predict measures of processing difficulty.

Recent studies in computational psycholinguistics have provided conflicting evidence with regard to the relationship between LM quality (i.e. next-word prediction accuracy) and goodness-of-fit to human reading times. Earlier work using newly-trained LMs showed a *negative* relationship between LM perplexity and predictive power of surprisal estimates (Goodkind and Bicknell, 2018; Wilcox et al., 2020; Merkx and Frank, 2021), but more recent work using large pre-trained Transformer-based LMs (e.g. GPT-2; Radford et al., 2019) show a robust *positive* relationship between the two variables (Oh et al., 2022; Oh and Schuler, 2023). While Oh and Schuler (2023) conjecture that these studies capture two distinct regimes, it remains less clear where the reversal in this relationship happens. The main challenge in answering this question lies in the massive difference in terms of both the amount of training data and the model capacity of LMs that were studied.

The current study aims to cover this conceptual middle ground by evaluating, on their ability to predict human reading times, surprisal estimates from Transformer-based LM variants that vary systematically in the amount of training data and model capacity. Results from regression analyses show that surprisal from most LM variants with contemporary model capacities make the biggest contribution to regression model fit after seeing about two billion tokens of training data, after which additional training data result in surprisal estimates that continue to diverge from humanlike expectations. Additionally, surprisal estimates from newly-trained smaller LM variants reveal a 'tipping point' at convergence, after which the decrease in perplexity begins to result in poorer fits to human reading times. Taken together, these results suggest that the vast amount of training data is mainly responsible for the poorer fit achieved by surprisal

from larger Transformer-based pre-trained LMs (Oh et al., 2022; Oh and Schuler, 2023), and that a certain degree of model capacity is necessary for Transformer-based LMs to capture humanlike expectations that manifest in reading times.

## 2 Experiment 1: Influence of Training Data Size

The first experiment examines the influence of training data size on the predictive power of Transformer-based LM surprisal by evaluating LM variants at various points in training on self-paced reading times from the Natural Stories Corpus (Futrell et al., 2021) and go-past eye-gaze durations from the Dundee Corpus (Kennedy et al., 2003).

### 2.1 Response Data

The Natural Stories Corpus contains reading times from 181 subjects that read 10 naturalistic English stories consisting a total of 10,245 tokens. The data points were filtered to remove those for sentence-initial and final words, those from subjects who answered three or fewer comprehension questions correctly, and those shorter than 100 ms or longer than 3000 ms, which resulted in a total of 384,905 observations in the exploratory set. The Dundee Corpus contains eye-gaze durations from 10 subjects that read 67 newspaper editorials consisting a total of 51,501 tokens. The data points were filtered to exclude those for unfixated words, words following saccades longer than four words, and sentence-, screen-, document-, and line-initial and final words, which resulted in a total of 98,115 observations in the exploratory set.[1] All observations were log-transformed prior to model fitting.

### 2.2 Predictors

This experiment evaluates surprisal estimates from eight variants of Pythia LMs (Biderman et al., 2023), whose intermediate parameters were saved at various points during training. Pythia LMs are decoder-only autoregressive Transformer-based models[2] whose variants differ primarily in their capacity. The model capacities of the Pythia variants are summarized in Table 1.

| Model | #L | #H | $d_{\mathrm{model}}$ | #Parameters |
|---|---|---|---|---|
| Pythia 70M | 6 | 8 | 512 | ∼70M |
| Pythia 160M | 12 | 12 | 768 | ∼160M |
| Pythia 410M | 24 | 16 | 1024 | ∼410M |
| Pythia 1B | 16 | 8 | 2048 | ∼1B |
| Pythia 1.4B | 24 | 16 | 2048 | ∼1.4B |
| Pythia 2.8B | 32 | 32 | 2560 | ∼2.8B |
| Pythia 6.9B | 32 | 32 | 4096 | ∼6.9B |
| Pythia 12B | 36 | 40 | 5120 | ∼12B |

Table 1: Model capacities of Pythia variants whose surprisal estimates were examined in this work. #L, #H, and $d_{\mathrm{model}}$ refer to number of layers, number of attention heads per layer, and embedding size, respectively.

Crucially for this experiment, all eight Pythia variants were trained using identical batches of training examples that were presented in the same order. These training examples come from the Pile (Gao et al., 2020), which is a collection of English language datasets that consist of around 300 billion tokens. Batches of 1,024 examples with a sequence length of 2,048 (i.e. 2,097,152 tokens) were used to train the eight variants for 143,000 steps, which amounts to about one epoch of the entire Pile dataset. Model parameters that were saved during early training stages (i.e. after 1, 2, 4, ..., 256, 512 steps) as well as after every 1,000 steps are publicly available.

Each article of the Natural Stories Corpus and each article of the Dundee Corpus was tokenized by Pythia's byte-pair encoding (BPE; Sennrich et al., 2016) tokenizer and provided as input to each model variant. For each model variant, all publicly available intermediate model weights were used to calculate surprisal estimates on the two corpora. In cases where each story or article was longer than a single context window of 2,048 tokens, surprisal estimates for the remaining tokens were calculated by using the second half of the previous context window as the first half of a new context window.

### 2.3 Regression Modeling

Subsequently, following previous work (Oh et al., 2022; Oh and Schuler, 2023), a 'baseline' linear mixed-effects (LME) model that contains baseline predictors for low-level cognitive processing, and 'full' LME models that additionally contain each LM surprisal predictor, were fit to self-paced reading times and go-past durations using lme4 (Bates et al., 2015). These baseline predictors are word length in characters and index of word position in each sentence (Natural Stories and Dundee), as well as saccade length and whether or not the pre-

---

[1] The held-out set of each corpus, which have a comparable number of observations, is reserved for statistical significance testing and therefore was not analyzed in this work.

[2] Technical details such as the parallelization of self-attention/feedforward computations and the separation of embedding/projection matrices differentiate Pythia LMs from other large language model families.

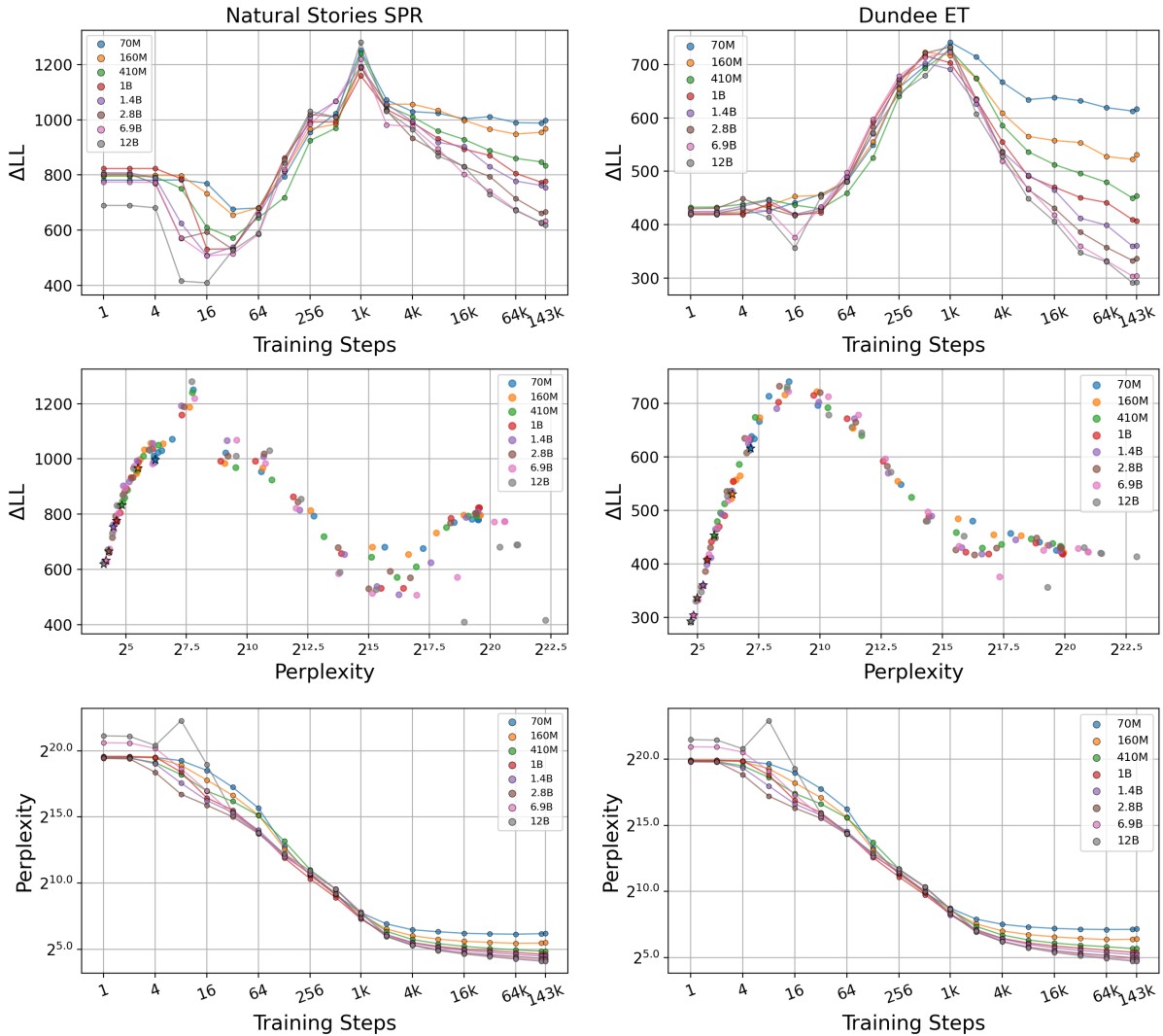

Figure 1: Increase in regression model log-likelihood due to including each surprisal estimate from Pythia variants as a function of training steps (top) and perplexity (middle; the stars indicate the fully trained versions after 143,000 steps), as well as perplexity as a function of training steps (bottom) on the exploratory set of Natural Stories (left) and Dundee data (right).

vious word was fixated (Dundee only). All predictors were centered and scaled,[3] and the LME models included by-subject random slopes for all fixed effects and random intercepts for each subject. In addition, a random intercept for each subject-sentence interaction was included for self-paced reading times collected from 181 subjects, and a random intercept for each sentence was included for eye-gaze durations collected from a smaller number of 10 subjects. Once the regression models were fit, the increase in regression model log-likelihood ($\Delta$LL) was calculated for each regression model by subtracting the log-likelihood of the baseline regression model from that of a full re-

gression model. Finally, the perplexity of each LM variant was calculated on the two corpora.

## 2.4  Results

The results in Figure 1 show that across both corpora, surprisal from most LM variants made the biggest contribution to regression model fit after 1,000 training steps (i.e. after about two billion tokens).[4] This seems to represent a 'humanlike

---

[3]'Spillover' predictors were not included in the regression models to avoid convergence issues.

[4]Results from after 2,000 steps were selectively omitted for clarity, as they were consistent with the general trend. As pointed out by a reviewer, including a frequency-based predictor in the regression models may change the exact location of this peak. However, this work avoids potential confounds introduced by the corpus used for frequency estimation by evaluating surprisal estimates on their own following the protocols of Oh and Schuler (2023).

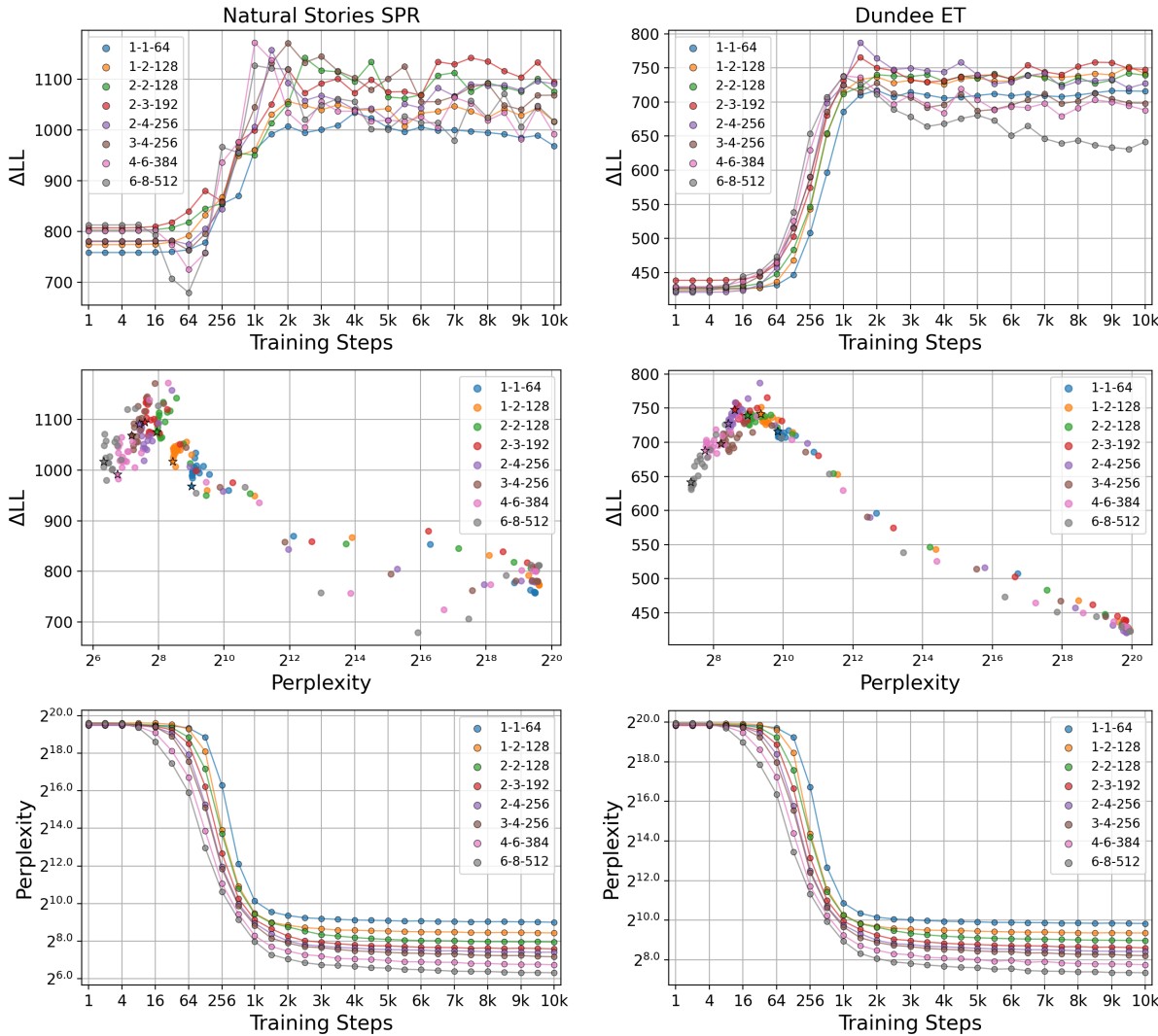

Figure 2: Increase in regression model log-likelihood due to including each surprisal estimate from newly-trained LM variants as a function of training steps (top) and perplexity (middle; the stars indicate the fully trained versions after 10,000 steps), as well as perplexity as a function of training steps (bottom) on the exploratory set of Natural Stories (left) and Dundee data (right). The variants are labeled using their number of layers, number of attention heads per layer, and embedding size, in that order.

optimum,' after which surprisal estimates begin to diverge from humanlike expectations as training continues. At this point in training, there appears to be no systematic relationship between model capacity and predictive power of surprisal estimates. However, after all 143,000 training steps (i.e. after about 300 billion tokens), the eight model variants show a strictly monotonic and negative relationship, which directly replicates the findings of Oh and Schuler (2023).[5] Taken together, these results indicate that the vast amount of training data is responsible for the poorer fit achieved by surprisal

from larger Transformer-based LMs.

# 3 Experiment 2: Influence of Model Capacity

The second experiment further examines the relationship between model capacity and predictive power of surprisal estimates by evaluating Transformer-based LM variants smaller than the Pythia variants at various points in training, following similar procedures as Experiment 1.

## 3.1 Procedures

Surprisal estimates from eight smaller LM variants were evaluated at various points during training in this experiment. The largest of these vari-

---

[5]The best-fitting line between log perplexity and $\Delta$LL of these variants had a slope significantly greater than 0 at $p < 0.05$ level according to a one-tailed $t$-test on both corpora.

ants has the same model capacity as the smallest Pythia 70M variant, and the smaller variants were designed to have fewer layers and attention heads, as well as smaller embeddings. These variants were trained closely following the training procedures of the Pythia variants, including the size and order of training batches. For computational efficiency, these variants were trained for the first 10,000 training steps, based on the observation that $\Delta$LL on both corpora did not change substantially after 8,000 steps for the smallest Pythia variant.[6] The predictive power of resulting surprisal estimates was evaluated following identical procedures as Experiment 1.

## 3.2 Results

The results in Figure 2 show that surprisal from the two largest variants made the biggest contribution to regression model fit after 1,000 training steps on both corpora, replicating the results of Experiment 1. In contrast, the smaller variants such as the 2-2-128 and 2-3-192 variants seem to peak later at around 2,000 training steps and stabilize afterward. After all 10,000 training steps, the model variants show a reversal in the relationship between LM perplexity and fit to reading times; the 2-3-192 variant seems to represent a 'tipping point,' after which the decrease in perplexity starts to result in poorer fits to human reading times. Additionally, variants that are smaller than this yield surprisal estimates that are less predictive of reading times when sufficiently trained. These results suggest that a certain degree of model capacity is necessary for Transformer-based LMs to capture humanlike expectations that manifest in reading times.

## 4 Discussion and Conclusion

This work aims to consolidate conflicting findings about the relationship between LM quality and the predictive power of its surprisal estimates by systematically manipulating the amount of training data and model capacity. Experimental results show that surprisal from most contemporary Transformer-based LM variants provide the best fit to human reading times with about two billion training tokens, after which they begin to diverge from humanlike expectations. It is conjectured that early training data up to about two billion tokens is

helpful for learning e.g. selectional preferences that align well with humanlike prediction and processing difficulty. However, as the models see more training data, they are able to achieve 'superhuman' prediction, which makes their surprisal estimates diverge more and more from human reading times as training continues. The words for which prediction by LMs improves with massive amounts of training data are likely to be open-class words like nouns and adjectives, whose reading times were identifed as being most severely underpredicted by their surprisal estimates (Oh and Schuler, 2023).

Moreover, at the end of training, these model variants show a strictly monotonic and negative relationship between perplexity and fit to human reading times. This directly replicates the findings of Oh et al. (2022) and adds to a growing body of research reporting an inverse correlation between model size and regression model fit (Kuribayashi et al., 2022; Shain et al., 2022; de Varda and Marelli, 2023). The current results demonstrate that this relationship emerges with large amounts of training data and becomes stronger as training continues. The bottleneck posed by the limited model capacity of the smaller variants appears to prevent them from learning to make excessively accurate predictions that cause the divergence between surprisal and human reading times. However, newly-trained LM variants that are smaller than those of contemporary standards reveal a 'tipping point' at convergence, which indicates that a certain amount of model capacity is necessary for LMs to correctly learn humanlike expectations.

Finally, across both experiments, model capacity does not seem to modulate the relationship between perplexity and fit to human reading times, with data points from different LM variants forming a continuous curve between log perplexity and $\Delta$LL. This suggests that Transformer-based LMs of different capacities share a similar inductive bias that initially improves the fit of surprisal estimates to human reading times but begins to have an adverse effect on it with large amounts of training data.

## Acknowledgments

We thank the reviewers and the area chair for their helpful comments. This work was supported by the National Science Foundation grant #1816891. All views expressed are those of the authors and do not necessarily reflect the views of the National Science Foundation.

---

[6]Refer to Appendix A for the model capacities of these variants as well as further details on their training procedures. Code and trained weights are available at `https://github.com/byungdoh/slm_surprisal`.

## Limitations

The connection between conditional probabilities of Transformer-based language models and human sentence processing drawn in this work is based on language model variants trained on English text and data from human subjects that are native speakers of English. Therefore, the connection made in this work may not generalize to other languages.

## Ethics Statement

Experiments presented in this work used datasets from previously published research (Futrell et al., 2021; Kennedy et al., 2003), in which the procedures for data collection and validation are outlined. As this work focuses on studying the connection between conditional probabilities of language models and human sentence processing, its potential negative impacts on society seem to be minimal.

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

## A    Model Capacities and Training Procedures of Smaller LM Variants

The eight LM variants that were trained as part of Experiment 2 are decoder-only autoregressive Transformer-based models that share the same architecture as the Pythia LM variants (Biderman et al., 2023). Their model capacities are summarized in Table 2.

These variants were trained using the GPT-NeoX library (Andonian et al., 2021) closely following the training procedures of the Pythia LM variants.[7] Identical training batches of 1,024 examples with a sequence length of 2,048 from the Pile (Gao et al.,

| Model | #L | #H | $d_{model}$ | #Parameters |
|---|---|---|---|---|
| Repro 1-1-64 | 1 | 1 | 64 | ∼6M |
| Repro 1-2-128 | 1 | 2 | 128 | ∼13M |
| Repro 2-2-128 | 2 | 2 | 128 | ∼13M |
| Repro 2-3-192 | 2 | 3 | 192 | ∼20M |
| Repro 2-4-256 | 2 | 4 | 256 | ∼27M |
| Repro 3-4-256 | 3 | 4 | 256 | ∼28M |
| Repro 4-6-384 | 4 | 6 | 384 | ∼46M |
| Repro 6-8-512 | 6 | 8 | 512 | ∼70M |

Table 2: Model capacities of newly-trained LM variants whose surprisal estimates were examined in this work. #L, #H, and $d_{model}$ refer to number of layers, number of attention heads, and embedding size, respectively.

2020) were provided to each variant in the same order as the Pythia variants. The variants were trained using the Zero Redundancy Optimizer (ZeRO; Rajbhandari et al., 2020) implementation of Adam (Kingma and Ba, 2015) with a learning rate of 0.001. The learning rate was warmed up linearly over the first 1% of training steps (i.e. 1,430 steps) and were subsequently lowered to a minimum of 0.0001 following a cosine annealing schedule over the remainder of the 143,000 training steps. However, for computational efficiency, training was stopped after the first 10,000 training steps. For comparability with the Pythia variants, intermediate parameters were saved during early training stages (i.e. after 1, 2, 4, ..., 256, 512 steps) as well as after every 500 steps from step 1,000 onward.

---

[7]The only minor difference is that the FlashAttention (Dao et al., 2022) implementation of scaled dot-product attention could not be used during training due to a mismatch in GPU hardware specifications.