# OpenReview forum: "Transformer-Based Language Model Surprisal Predicts Human Reading Times Best with About Two Billion Training Tokens"
_EMNLP/2023/Conference — EMNLP 2023 Findings_

### Official Review · Reviewer_AxD9 · 2023-08-02

**Soundness:** 4

**Excitement:**

4: Strong: This paper deepens the understanding of some phenomenon or lowers the barriers to an existing research direction.

**Missing References:**

https://aclanthology.org/2021.acl-long.405/
https://aclanthology.org/2022.emnlp-main.712/

**Paper Topic And Main Contributions:**

This paper aims to empirically investigate the relationship between language model surprisals and human reading times and in particular the relationship between model capacity and training set size and model performance in predicting RT. The paper aims to elucidate the mysterious findings that a positive relationship between LM quality and RT predictive capacity seems to reverse for very large models. The paper finds that the reversal happens at around 1k training steps, consistently for two datasets in two different modalities (eye tracking vs. self-paced reading).

**Reasons To Accept:**

- The consistency of the results is striking given the different nature of the datasets; the datasets have very different genres of text and were collected under very different circumstances. This increases my credence that the result is robust.

- The paper makes a nice contribution to the literature and support Oh and Schuler’s idea that the reason very large language models don’t model human reading times well is that they are “too good” at prediction, and know too many things such as low-frequency entity names that humans do not know. In some ways the paper raises more questions than it answers (why the inflection point at 1k?) but this is good as it will inspire future work.

**Reasons To Reject:**

- The paper claims that “a certain degree of model capacity is necessary for Transformer-based LMs to capture humanlike expectations that manifest in reading times.” It’s not clear to me from the text, nor from the figures, what exactly the claim is here. Is it that models need to have low capacity to model human reading times, and too-high capacity is deleterious? If so would that support accounts like Futrell, Gibson & Levy’s (2020) lossy-context surprisal? Or is it that a model with too high capacity can learn too much from the training data? Or is the claim that models need a certain capacity and beyond this there are diminishing returns? It’s not clear to me what the specific claim here is.

- I also think the paper could be improved. First, some details of the analysis aren’t clear. Was spillover accounted for and how? It would also be nice to move beyond the Dundee and NS datasets. Also, in terms of the analysis, it would be interesting to see plots comparing Delta LL for the different model architectures in Figure 2 but matched in perplexity (by using earlier checkpoints). This could answer questions about the real influence of model capacity.


**Reproducibility:**

5: Could easily reproduce the results.

**Reviewer Confidence:**

5: Positive that my evaluation is correct. I read the paper very carefully and I am very familiar with related work.

---

> ### Author Rebuttal · Authors · 2023-08-28
>
> Thank you for your critical assessment and constructive feedback on this work, as well as references to earlier work that observe a similar relationship between language model perplexity and fit to human reading times. We will situate our work in relation to these results in a “Discussion” section if the paper is accepted.
>
> According to the guidelines for author response, below we address your concerns and questions point by point, presenting new results where appropriate.
>
> - **Reasons to Reject #1:** We are sorry that the interpretation of the results was not clear for the two experiments. The first experiment shows that surprisal from larger language models begins to demonstrate a poorer fit to reading times only after seeing a certain amount of training data (i.e. ~2 billion tokens), which suggests that larger language models can learn to achieve ‘superhuman’ prediction that deviates from humanlike processing difficulty (Oh & Schuler, 2023, “Why does surprisal from larger Transformer-based language models provide a poorer fit to human reading times?”) given the same amount of training data as the smaller language models. However, the second experiment shows that models that are too small are not able to learn enough from the training data, with models that are smaller than the ‘2-3-192’ variant yielding surprisal estimates that are less predictive of human reading times when sufficiently trained. We will add a discussion of these points in a “Discussion” section if the paper is accepted.
>
> - **Reasons to Reject #2:** We will include additional details of the regression analysis. In our experiments, spillover was not controlled for, following the protocols of Oh and Schuler (2023) in order to avoid issues with regression model identifiability.
>
> - **Reasons to Reject #2:** We additionally conducted regression analyses on two more eye-tracking datasets, namely the Ghent Eye-tracking Corpus (Cop et al., 2017, “Presenting GECO: An eyetracking corpus of monolingual and bilingual sentence reading”) and the Provo Corpus (Luke & Christianson, 2018, “The Provo Corpus: A large eye-tracking corpus with predictability norms”), following the same procedures as the first experiment. The two main results (i.e. the peak in ΔLL at 1k training steps, poorer fit to reading times by surprisal from larger language models at the end of training) were robustly replicated. We will include these results in the Appendix if the paper is accepted.
>
> - **Reasons to Reject #2:** Generally, model capacity does not seem to modulate the relationship between perplexity and ΔLL; the bottom panels of Figure 1 show that data points from models with different capacities form one continuous curve between log perplexity and ΔLL. We will additionally include a description of this observation.

---

### Official Review · Reviewer_vi7K · 2023-08-04

**Soundness:** 4

**Excitement:**

3: Ambivalent: It has merits (e.g., it reports state-of-the-art results, the idea is nice), but there are key weaknesses (e.g., it describes incremental work), and it can significantly benefit from another round of revision. However, I won't object to accepting it if my co-reviewers champion it.

**Paper Topic And Main Contributions:**

This paper presents several experiments to account for a dissociation in the prediction of human behavioral data -- some models improve in their psychometric fit as they get better at modeling linguistic regularities, while others deviate. The paper parametrically manipulates the size of the training data and demonstrates that there is a consistent peak around 2 billion tokens in being able to account for reading times. It suggests that the apparent discrepancy in earlier modeling work is driven by training dataset size.

**Questions For The Authors:**

A. It would be nice to see a plot of perplexity by training steps - Acknowledged
B. I would also like to see R^2 reported as well as significance tests for those sections where fit is assessed -- currently it is only described qualitatively and it would be nice to know that the curves are in fact distinct and/or that there are significant differences as a function of model parameters - Acknowledged

**Reasons To Accept:**

* Paper is easy to read
* Experiments are well-described and soundly conducted
* Evaluation is fair and conducted over multiple reading time datasets

**Reasons To Reject:**

* It is not clear what this answers with respect to any cognitive questions - Author response indicates a restatement of the introduction which does not answer a cognitive question (rather an empirical one); restatement could be clearer in the body of the text if this is truly a sufficient response to my question.
* The specific mechanisms about the psychometric fit and where it peaks are not well-understood - The authors do not address the theoretical questions outlined by "why is it these specific values." The speculations in the text are critical to their scientific question -- it is not trivial to raise one's hands up about the theory.
* Results are limited to the Pythia models - The author response is sufficient. This is however a limitation about the generality of these effects that could be made clear earlier.
* Surprisal is assumed to be the linking function which does not fully address the "psychometric fit" issue - The authors make an earnest attempt at a response. However, my concern is that we are too focused on using surprisal to explain reading times. If one disagrees with the premise, that makes this experiment superfluous. Thus, the authors' response to my concern also belongs in the text.
* Eyetracking data selection/filtering is a bit perplexing -- it looks like everything is analyzed (e.g., regressions to earlier parts of the sentence) with some exclusions. - Thank you for the clarification. Please place this information more prominently in the text.

I am particularly worried that the work is a bit "atheoretical" with respect to psycholinguistic phenomena. I don't really understand why the peak in psychometric fit occurs where it does and the paper's explanations do not address our understanding of language modeling (memorization, the source of deviations, etc.), our understanding of surprisal, or relationships between context and language comprehension.

**Reproducibility:**

3: Could reproduce the results with some difficulty. The settings of parameters are underspecified or subjectively determined; the training/evaluation data are not widely available.

**Reviewer Confidence:**

5: Positive that my evaluation is correct. I read the paper very carefully and I am very familiar with related work.

**Typos Grammar Style And Presentation Improvements:**

* "LM" in the title should be spelled out to state "Language Model"

---

> ### Author Rebuttal · Authors · 2023-08-28
>
> Thank you for your critical assessment and constructive feedback on this work. Some of the confusion may be due to the fact that the discussion of the results was abrogated due to space constraints. We will include a discussion of the points below in a “Discussion” section in the allotted extra page if the paper is accepted.
>
> According to the guidelines for author response, below we address your concerns and questions point by point, presenting new results where appropriate.
>
> - **Reasons to Reject #1:** The main theoretical implication of this work is that the relationship between next-word prediction accuracy (i.e. perplexity) and the predictive power of surprisal estimates is modulated by the amount of training data language models have seen. Until very recently, it was thought that more accurate language models are better approximators of humanlike expectations, and this assumption has even informed recent work on the functional form between surprisal and reading times (Hoover et al., 2023, “The plausibility of sampling as an algorithmic theory of sentence processing”). Our work shows that after about two billion training tokens, more accurate predictions lead to divergence from humanlike expectations.
>
> - **Reasons to Reject #2:** With regard to the peak observed in this work, we speculate that training data up to two billion tokens is helpful for learning e.g. selectional preferences that align well with humanlike expectations and processing difficulty. However, as the models see more training data, the models are able to achieve ‘superhuman’ levels of prediction, which make their surprisal estimates diverge more and more from human reading times as training continues. This is consistent with the results of Oh and Schuler (2023, “Why does surprisal from larger Transformer-based language models provide a poorer fit to human reading times?”), who found that excessively accurate predictions of open-class words like proper nouns and adjectives lead to underpredictions of human reading times.
>
> - **Reasons to Reject #3:** To our knowledge, the Pythia models are the only pre-trained language models that use purely publicly available training data and have publicly available checkpoints at various points during training.
>
> - **Reasons to Reject #4:** While this work evaluates language models under the assumptions of surprisal theory, it is less straightforward to evaluate them under other (e.g. memory-based) theoretical frameworks, as their only training objective is to predict the next word. Additionally, the fact that surprisal from these language models is predictive of human reading times across two corpora and various model sizes (i.e. all ΔLL are positive and substantial) provides support for this framework. The ability of language model surprisal to predict measures of comprehension difficulty has been replicated many times in recent work (Wilcox et al., 2020, “On the predictive power of neural language models for human real-time comprehension behavior”; Shain et al., 2020, “fMRI reveals language-specific predictive coding during naturalistic sentence comprehension”, inter alia).
>
> - **Reasons to Reject #5:** Go-past durations were analyzed in this work as regressions are thought to reflect additional processing difficulty that is incurred by the current word. We additionally conducted regression analyses using the scan-path fixation durations of the Dundee corpus instead, but the overall findings (i.e. the peak in ΔLL at 1k training steps, poorer fit to reading times by surprisal from larger language models at the end of training) did not change.
>
> - **Question A:** Perplexity decreases monotonically as a function of training steps, with the total decrease being larger for the larger language model variants at the end of training. We will include a visualization of these two variables in the Appendix if the paper is accepted.
>
> - **Question B:** We will include a visualization of the regression models’ R$^2$ values. Additionally, we will report the results of a non-parametric permutation test that tests whether the slope between log perplexity and ΔLL is greater than that expected by chance at the end of language model training (the stars in the bottom panels of Figure 1; *p* < 0.001 on both corpora).

---

### Official Review · Reviewer_nrTc · 2023-08-06

**Soundness:** 3

**Excitement:**

3: Ambivalent: It has merits (e.g., it reports state-of-the-art results, the idea is nice), but there are key weaknesses (e.g., it describes incremental work), and it can significantly benefit from another round of revision. However, I won't object to accepting it if my co-reviewers champion it.

**Missing References:**

- Tatsuki Kuribayashi, Yohei Oseki, Takumi Ito, Ryo Yoshida, Masayuki Asahara, and Kentaro Inui. 2021. Lower perplexity is not always human-like. ACL.

**Paper Topic And Main Contributions:**

The main contribution of this paper is to demonstrate that the correlation between model quality and psychometric predictive power does hold beyond two billion training tokens.

**Questions For The Authors:**

- A: Why does "the 2-3-192 variant" represent "a tipping point"?
- B: How are random-effects structures of regression modeling determined for self-paced reading times and eye-gaze durations, respectively?

**Reasons To Accept:**

- To the best of my knowledge, this paper is the first systematic investigation of the correlation between model quality and psychometric predictive power with respect to training data and model capacity.

**Reasons To Reject:**

- While the results of training data are clear, the results of model capacity are hard to interpret. For example, the observation that "the smaller variants seem to peak later at around 2,000 training steps and stabilize afterward" cannot be seen from Figure 2.
- The reason why two billion training tokens are the magical number remains to be discussed.

**Reproducibility:**

1: Could not reproduce the results here no matter how hard they tried.

**Reviewer Confidence:**

5: Positive that my evaluation is correct. I read the paper very carefully and I am very familiar with related work.

---

> ### Author Rebuttal · Authors · 2023-08-28
>
> Thank you for your critical assessment and constructive feedback on this work, as well as references to earlier work that observed a similar relationship between language model perplexity and fit to human reading times on Japanese data. We will situate our work in relation to these results in a “Discussion” section if the paper is accepted.
>
> According to the guidelines for author response, below we address your concerns and questions point by point, presenting new results where appropriate.
>
> - **Reasons to Reject #1:** We are sorry that the description of the results was not clear for the experiment manipulating model capacity. The observation that “the smaller variants seem to peak later at around 2,000 training steps and stabilize afterward” was meant to describe that unlike the two largest variants examined in the experiment (i.e. ‘4-6-384’ and ‘6-8-512’ variants), the smaller variants (e.g.  ‘2-2-128’, ‘2-3-192’, and ‘2-4-256’ variants) have a later peak and do not demonstrate a substantial decrease in ΔLL by training step 10k, suggesting their convergence. This can also be seen in the bottom panels of Figure 2, where perplexity stops decreasing for the smaller variants (resulting in clusters of data points). We will clarify this description if the paper is accepted.
>
> - **Reasons to Reject #2:** We speculate that training data up to two billion tokens is helpful for learning e.g. selectional preferences that align well with humanlike prediction and processing difficulty. However, as the models see more training data, they are able to achieve ‘superhuman’ prediction, which makes their surprisal estimates diverge more and more from human reading times as training continues. This is consistent with the findings of Oh and Schuler (2023, “Why does surprisal from larger Transformer-based language models provide a poorer fit to human reading times?”), and we will add a discussion of these points in a “Discussion” section if the paper is accepted.
>
> - **Question A:** Based on the results of the first experiment, a natural question to ask is how much smaller Transformer language models can become before their surprisal estimates begin to degrade. In this regard, the ‘2-3-192’ variant (the red stars in the bottom panels of Figure 2) represents a ‘tipping point’ in the sense that language models that are smaller in scale begin to yield surprisal estimates that are less predictive of human reading times when sufficiently trained.
>
> - **Question B:** We followed the experimental protocols of Oh and Schuler (2023) to determine the random effects structures on the two corpora of human reading times. Their protocol is consistent with the conventional practice of psycholinguistics of keeping maximal by-subject random effects (Barr et al., 2013, “Random effects structure for confirmatory hypothesis testing: Keep it maximal”).
>
> - **Reproducibility:** With regard to your concern for reproducibility, we would like to point out that the pre-trained Pythia models that were examined in the first experiment as well as the training data for models examined in the second experiment are publicly available (Biderman et al., 2023, “Pythia: A suite for analyzing large language models across training and scaling”), as well as the regression software and the Natural Stories Corpus. We also plan to publicly release weights of all newly-trained models as well as code used for surprisal calculation.

---

### Meta-Review · Area_Chair_u9Pe · 2023-09-19

**Recommendation:** 4

**Metareview:**

# Overview

Prior work in psycholinguistics has shown that as language models become better (in terms of perplexity/cross-entropy) their surprisal estimate’s ability to predict reading times (RTs) improves (Goodkind et al. 2018; Wilcox et al. 2020).
A few recent papers, however, show this trend reverses after a certain “language model quality threshold” (i.e., GPT-2 small’s surprisals have better predictive power than either GPT-2 XL or GPT-3; Oh et al. 2022; Oh and Schuller 2023; Shain et al. 2022).
This paper investigates when this trend-reversal occurs, by analysing how a model’s psychometric predictive power changes over training—using a number of language models with different sizes.
They find clear results suggesting that (under their experimental conditions) a language model’s psychometric predictive power peaks when they are exposed to roughly two billion training tokens.

# Meta review

The reviewers generally agree that this paper is well-written, sound and makes an important contribution to the literature. Further, the fact that experiments are consistent across a range of datasets and RT measurements is both striking and reassuring. Personally, I believe that if these results are confirmed this could be a very impactful paper (but see my personal review below).

The reviewers also point out some issues with the paper. In particular, the paper does not try to address the question of “why two billion training tokens?”---i.e., it does not try to address the cognitive questions behind this observed trend. Further, the impact of model capacity on the observed trends seems unclear. Finally, the authors claim they did not include spillover effects in their analysis due to issues with regression model identifiability, but: (i) including spillover effects is common practice and these effects can be stronger than the main effects depending on the dataset (in e.g., Brown); (ii) I do not see why identifiability should be relevant to this particular study which focuses on delta log-likelihoods (a longer form justification could be useful). In my opinion, though, these issues are either addressable for camera ready, or not large enough to warrant rejection.

# My Review

First, I'd like to say that I agree with the positive points about the paper summarised above.
However, I personally believe that this paper has a significant experimental flaw: it **does not seem to control for unigram log-probabilities** in its analyses.

This may be particularly problematic because:

* It is well-known that unigram log-probabilities have a strong predictive power over reading times.
Beyond the most recent confirmation of the effect (in Shain, 2023, which was released after EMNLP’s CFP), most prior work in this vein controlled for frequency effects (Goodkind et al. 2018; Wilcox et al. 2020, 2023; Oh et al. 2022; Shain et al. 2022).
In particular, a close look at Oh et al.’s (2022) results (comparing fig. 5 to 7 or fig. 6 to 8) shows that the predictability of unigram log-probability over RTs is larger than that of surprisal for these experimental conditions.

* Figure 5 in Chang and Bergen (2022) shows that a language model’s distribution (and by proxy its surprisals) approximate the unigram distribution most closely after about 1000 training updates (also replicated in Meister et al. 2023).

Together, these results suggest that the effect demonstrated in this paper is that: transformers approximate unigram at roughly 1k training steps; and unigram log-probabilities predict RTs better than language model’s surprisals.
In conclusion, by not controlling for frequency effects this paper might be confounding these two trends.

Furthermore, prior work (Goodkind et al. 2018; Wilcox et al. 2020) which showed that *improving language models improves their psychometric predictive power over reading times* has always controlled for frequency in their analysis.
More recent work which shows this trend stops when models get larger than GPT-2 small (Oh et al. 2022; Shain et al. 2022) also controlled for frequency.
To me, by not controlling for frequency, this paper might be investigating a different phenomenon.
In case this paper is accepted, I would recommend adding new experiments addressing this issue (or at least a disclaimer about this alternative interpretation of the results).

# References

Shain et al. 2022. Large-Scale Evidence for Logarithmic Effects of Word Predictability on Reading Time

Shain. 2023. Word Frequency and Predictability Dissociate in Naturalistic Reading

Goodkind et al. 2018. Predictive power of word surprisal for reading times is a linear function of language model quality

Wilcox et al. 2020. On the Predictive Power of Neural Language Models for Human Real-TimeComprehension Behavior

Wilcox et al. 2023. Testing the Predictions of Surprisal Theory in 11 Languages

Chang and Bergen. 2022. Word Acquisition in Neural Language Models

Meister et al. 2023. A Natural Bias for Language Generation Models

---

### Decision · Program_Chairs · 2023-10-07

**Decision:**

Accept-Findings

**Comment:**

# Overview

Prior work in psycholinguistics has shown that as language models become better (in terms of perplexity/cross-entropy) their surprisal estimate’s ability to predict reading times (RTs) improves (Goodkind et al. 2018; Wilcox et al. 2020).
A few recent papers, however, show this trend reverses after a certain “language model quality threshold” (i.e., GPT-2 small’s surprisals have better predictive power than either GPT-2 XL or GPT-3; Oh et al. 2022; Oh and Schuller 2023; Shain et al. 2022).
This paper investigates when this trend-reversal occurs, by analysing how a model’s psychometric predictive power changes over training—using a number of language models with different sizes.
They find clear results suggesting that (under their experimental conditions) a language model’s psychometric predictive power peaks when they are exposed to roughly two billion training tokens.

# Meta review

The reviewers generally agree that this paper is well-written, sound and makes an important contribution to the literature. Further, the fact that experiments are consistent across a range of datasets and RT measurements is both striking and reassuring. Personally, I believe that if these results are confirmed this could be a very impactful paper (but see my personal review below).

The reviewers also point out some issues with the paper. In particular, the paper does not try to address the question of “why two billion training tokens?”---i.e., it does not try to address the cognitive questions behind this observed trend. Further, the impact of model capacity on the observed trends seems unclear. Finally, the authors claim they did not include spillover effects in their analysis due to issues with regression model identifiability, but: (i) including spillover effects is common practice and these effects can be stronger than the main effects depending on the dataset (in e.g., Brown); (ii) I do not see why identifiability should be relevant to this particular study which focuses on delta log-likelihoods (a longer form justification could be useful). In my opinion, though, these issues are either addressable for camera ready, or not large enough to warrant rejection.

# My Review

First, I'd like to say that I agree with the positive points about the paper summarised above.
However, I personally believe that this paper has a significant experimental flaw: it **does not seem to control for unigram log-probabilities** in its analyses.

This may be particularly problematic because:

* It is well-known that unigram log-probabilities have a strong predictive power over reading times.
Beyond the most recent confirmation of the effect (in Shain, 2023, which was released after EMNLP’s CFP), most prior work in this vein controlled for frequency effects (Goodkind et al. 2018; Wilcox et al. 2020, 2023; Oh et al. 2022; Shain et al. 2022).
In particular, a close look at Oh et al.’s (2022) results (comparing fig. 5 to 7 or fig. 6 to 8) shows that the predictability of unigram log-probability over RTs is larger than that of surprisal for these experimental conditions.

* Figure 5 in Chang and Bergen (2022) shows that a language model’s distribution (and by proxy its surprisals) approximate the unigram distribution most closely after about 1000 training updates (also replicated in Meister et al. 2023).

Together, these results suggest that the effect demonstrated in this paper is that: transformers approximate unigram at roughly 1k training steps; and unigram log-probabilities predict RTs better than language model’s surprisals.
In conclusion, by not controlling for frequency effects this paper might be confounding these two trends.

Furthermore, prior work (Goodkind et al. 2018; Wilcox et al. 2020) which showed that *improving language models improves their psychometric predictive power over reading times* has always controlled for frequency in their analysis.
More recent work which shows this trend stops when models get larger than GPT-2 small (Oh et al. 2022; Shain et al. 2022) also controlled for frequency.
To me, by not controlling for frequency, this paper might be investigating a different phenomenon.
In case this paper is accepted, I would recommend adding new experiments addressing this issue (or at least a disclaimer about this alternative interpretation of the results).

# References

Shain et al. 2022. Large-Scale Evidence for Logarithmic Effects of Word Predictability on Reading Time

Shain. 2023. Word Frequency and Predictability Dissociate in Naturalistic Reading

Goodkind et al. 2018. Predictive power of word surprisal for reading times is a linear function of language model quality

Wilcox et al. 2020. On the Predictive Power of Neural Language Models for Human Real-TimeComprehension Behavior

Wilcox et al. 2023. Testing the Predictions of Surprisal Theory in 11 Languages

Chang and Bergen. 2022. Word Acquisition in Neural Language Models

Meister et al. 2023. A Natural Bias for Language Generation Models